# Research on Evaluation Method for Urban Water Circulation Health and Related Applications: A Case Study of Zhengzhou City, Henan Province

**DOI:** 10.3390/ijerph191710552

**Published:** 2022-08-24

**Authors:** Mengdie Zhao, Jinhang Li, Jinliang Zhang, Yuping Han, Runxiang Cao

**Affiliations:** 1College of Water Resources, North China University of Water Resources and Electric Power, Zhengzhou 450046, China; 2Yellow River Survey Planning and Design Institute Co., Ltd., Zhengzhou 450046, China

**Keywords:** healthy water cycle, combination weight, TOPSIS model, obstacle factor analysis, Zhengzhou City

## Abstract

The acceleration of urbanization and climate change has increasingly impacted the health level of urban dual water cycles. In order to accurately evaluate the health status of urban water cycles, the evaluation system covers four standard layers of water ecology, water abundance, water quality and water use, including 19 basic indicators such as water storage change and annual average precipitation. Three-scale AHP and EFAST algorithms are adopted to set the criterion and index layer weights. Water-cycle health assessment models are based on the improved TOPSIS model. The model evaluated Zhengzhou’s water cycle health from 2011 to 2021. We compared the TOPSIS model and FCE method to ensure the scientific objectivity of the evaluation results. The evaluation results indicated that the water cycle in Zhengzhou City improved annually, and the relative progress in 2020 was 0.567 in a sub-health state. The eco-environmental water demand, green coverage rate of the built district, water resources amount, and industry’s water consumption per unit of value added (CNY 10,000) were the major obstacles. These four factors have preponderantly influenced Zhengzhou City’s water cycle health. Our research results provide scientific reference for Zhengzhou to achieve a healthy urban water cycle and regional sustainable development.

## 1. Introduction

The water cycle supports the formation of water resources and is the main driving force for the evolution of water environments and ecosystems. With the development of the social economy and acceleration of urbanization and other human activities, the regional water cycle has gradually changed from a single “natural” water cycle model to a “natural–social” dual water cycle model. The basic process of a natural water cycle is precipitation–runoff–evaporation. The basic process of a social water cycle is water intake–water use–water supply–water drainage–water reuse. Natural and social water cycles influence each other and system structure coupling. A set of “natural-social” dual water cycle models based on “water supply–water use–water consumption–water drainage–water reuse” is formed.

Zhengzhou City is in the north-central part of Henan Province (Figure 1). The water cycle is greatly influenced by the urbanization rate and urbanization processes with problems including severe water pollution, a sharp decline in groundwater depth, and water shortage. In 2020, Zhengzhou’s total water resources were 859.12 million cubic metres. The per capita water resources were only 68 cubic metres, one-tenth of the national per capita share, far below the internationally recognized 500 cubic metres, signifying an extreme water shortage warning line. In general, Zhengzhou is in a state of severe water shortage. In recent years, the rapid economic development in Zhengzhou has led to increased water supply demand. In 2020, the total amount of sewage discharge reached 105.125 million tons, which restricted the long-term social development in Zhengzhou. Maintaining a healthy water cycle has become a significant proposition for the sustainable development of human society.

At the beginning of the 21st century, China’s urban water system developed rapidly. Many experts and scholars began to study the healthy circulation of urban water systems from the overall direction of the water environment and ecology. Zhang et al. [1] proposed a healthy water cycle according to the water crisis caused by human behaviours and emphasized the importance of recycling water models for water cycle health. Some scholars have adopted different methods to study this concept from different dimensions. For example, Jia et al. [2] evaluated the water cycle health in Ningxia and five other cities using principal component analysis. Zhang et al. [3] analysed the water cycle health in Beijing and comprehensively considered the influence of the South–North Water Diversion Project on the water cycle in Beijing, thus analysing the urban water cycle’s influencing factors more scientifically. Luan et al. [4] proposed a farmland healthy water cycle and evaluated the water cycle health of the Jun Liu irrigation area. Research on water cycle in foreign countries has obtained many research results. Chu [5] established an index system based on “S-E”, thought to evaluate the health status of the urban water cycle. Maria et al. [6] used life cycle methods to evaluate urban water cycles in the Mediterranean region. Deng et al. [7] proposed an improved entropy-based fuzzy matter element model to evaluate the Taihu Lake Basin in China. Healthy water cycles are a fuzzy concept.

The TOPSIS model ignores the linear relationship between indicators, leading to the failure of Euclidean geometry. In addition, Euclidean distance can only reflect the evaluation object and reason from position distance. The closeness of the solution cannot reflect similar geometry. Mahalanobis distance and grey relational analysis improve the TOPSIS method, which can evaluate healthy water circulation and provide references for water cycle health and ecological management in Zhengzhou.

In this context, Zhengzhou was chosen for case studies, and an evaluation system covering four criteria layers was established, namely water abundance, water ecology, water use, and water quality, including 19 evaluation indexes such as mean annual precipitation, annual water storage change of reservoir, and eco-environmental water demand. The combined weights of subjective and objective indexes were obtained by combining the three-scale AHP and EFAST algorithms. On this basis, we established an improved TOPSIS model to evaluate the water cycle in Zhengzhou. We used the TOPSIS model and FCE for comparison and verification, aiming to provide scientific reference for the realization of benign water cycles and sustainable development in Zhengzhou City.

## 2. Materials and Methods

### 2.1. Data Source

The data derive from China Environmental Statistics Yearbook (2010–2020), Zhengzhou City Water Resources Bulletin (2010–2020), Zhengzhou City Environmental Bulletin (2010–2020), Zhengzhou City Statistical Yearbook (2010–2020), Henan Province Water Resources Bulletin (2010–2020), Henan Province Statistical Yearbook (2010–2020), and so on.

### 2.2. The Connotation of Healthy Water Cycle

The traditional concept emphasizes that humans should not interfere with the natural water cycle to maintain its intrinsic natural state. However, with the intensification of human activities, the social water cycle’s role has gradually enhanced. Healthy water cycles should consider natural and human factors. On the one hand, the natural water cycle must ensure the social “three-living” water use under the premise of health and support the long-term development of society. On the other hand, human society must use water resources efficiently and avoid irreversible damage to the natural water cycle [8,9]. A healthy water cycle is manifested in the interaction between the natural and social water cycles on the cycle path, achieving dynamic balance. A healthy water cycle is demonstrated in normalizing water ecology, reasonably developing water resource abundance, healthy water environment quality, and efficient resource utilization [10,11]. Therefore, maintaining a healthy water cycle mainly includes the following three points: Firstly, water resource management must ensure the normalization of ecological water levels and self-healing capabilities. Secondly, the amount of water resources satisfies the need for socially sustainable development. Thirdly, efficiency in water resource utilization patterns [12]. A diagram of the urban healthy water cycle is shown in Figure 2.

### 2.3. Establishment of Evaluation Index System

The urban water cycle health assessment is a complex process. The natural and social water cycles should be included in the established evaluation index system. If the indicators are not comprehensive enough, the urban health status cannot be truly reflected. Therefore, the selection of indicators should follow objective laws and be regionally representative [13,14]. We comprehensively consider the connotation and regional status of water cycles in healthy cities. The relevant basic data of Zhengzhou City were collected by referring to the Water Resources Bulletin of Zhengzhou City and Water Resources Bulletin of Henan Province.

The health evaluation system of the water cycle was established from four dimensions: water supply, water sources, water use, and water treatment. The water utility criterion layer mainly determined the index layer from the effective utilization coefficient of irrigation water and social “three living”, which reflects the utilization efficiency of water resources in cities. The water abundance criterion layer mainly considered the natural endowment conditions of urban water resources, such as the annual water storage variation of medium-sized reservoirs and annual average precipitation. The water ecological criterion layer mainly established the index system from the human protection measures of water resources, river and lake storage capacity, and drainage pipeline density. The water quality reflected natural attributes of the water cycle and mainly involved urban greening and water function area status. The evaluation index system is shown in Figure 3.

#### Threshold Standard of Evaluation Index

In this paper, we determined the index threshold by combining the investigation and referring to relevant research results and national regulations to ensure scientific objectivity [15]. Due to the relativity of the water cycle’s health degree, the health standard was divided into five grades: excellent (grade I), healthy (grade II), sub-healthy (grade III), unhealthy (grade IV), and sick (grade V). The thresholds and attributes of each indicator in the evaluation system are shown in Table 1.

### 2.4. Research Method

#### 2.4.1. Three-Scale AHP–EFAST Algorithm

According to the principle of minimum information entropy, the three-scale AHP and EFAST algorithm were combined, and the subjectivity and objectivity of the weight were coupled. The weight of the evaluation index was accurate and reasonable. The calculation steps were as follows.
First item: The data were standardized.Since the annual torrential rain occurred in Zhengzhou City in 2021, the extreme value data were not representative. Therefore, we used a standardized transformation method to eliminate the contingency of extreme values.
(1)$$r_{\mathrm{ij}}=(x_{\mathrm{ij}}-\bar{X_{j}})/S_{j}$$
where $x_{\mathrm{ij}}$ is the original value of the evaluation index; $r_{\mathrm{ij}}$ is the index value after standard transformation and $\bar{X_{j}}$ is the mean value of item *j*; $S_{j}$ is the standard deviation of indicator *j*.Second item: Three-Scale AHP to Calculate Subjective WeightThe analytic hierarchy process (AHP) is a multi-objective decision analysis method that combines qualitative and quantitative analyses. AHP can realize the qualitative and quantitative determination of target weight, which is widely applicable for management and decision-making [16]. The main calculation process was divided into three steps:Establishment of Judgment Matrix *U*
$=\left(\begin{array}{ccc} U_{11} & \cdots & U_{1n}\\ \vdots & \ddots & \vdots \\ U_{n1} & \cdots & U_{\mathrm{nn}} \end{array}\right)$ In the matrix: $U_{\mathrm{ij}}$ is number 1 to 9 and its reciprocal, representing the relative influence between the two indexes. In this paper, we only used a scale of 0–2 to establish the judgment matrix, which improved the shortcomings of the traditional analytic hierarchy process.Calculate the maximum eigenvalue ($\lambda_{\max}$) of the judgment matrix:(2)$$\lambda_{\max}=\frac{1}{n}\sum_{i=1}^{n}\frac{{\left(\mathrm{Uw}\right)}_{i}}{w_{i}}$$
(3)$$w_{i}=\frac{\sqrt[{n}]{\prod_{j=1}^{n}U_{\mathrm{ij}}}}{\sum_{i=1}^{n}\sqrt[{n}]{\prod_{j=1}^{n}U_{\mathrm{ij}}}}$$Consistency test of calculation results. When C.R. was ≤0.1, the consistency test was passed. Otherwise, the matrix had to be adjusted until it was passed. The consistency test index formula for the judgment matrix is:(4)C.R.=C.I./R.I.R.I. is the random consistency index, which can be obtained from Table 2.
(5)$$C.I.=\frac{1}{n-1}{(\lambda }_{\max}-n)$$Third item: EFAST Algorithm to Calculate Objective Weight [17,18]The EFAST algorithm reflects sensitivity by the variance of the coupling between the indicators. The index weight with high sensitivity is higher, and the index weight with low sensitivity is lower. The algorithm is briefly introduced as follows:The total variance of model output can be represented as the sum of the variances of coupling effects with every index:(6)$$V=\sum_{i}V_{i}+\sum_{i\neq j}V_{ij}+\sum_{i\neq j\neq k}V_{ijk}+\ldots +\sum V_{ijk\ldots n}$$
where $V_{i}$ is the variance of index $x_{i}$; $V_{\mathrm{ij}}$ is the variance of the interaction between $x_{i}$ and $x_{j}$; $V_{\mathrm{ijk}}$ is the variance of the interaction between $x_{i}$, $x_{j}$ and $x_{k}$; $V_{\mathrm{ijk}\ldots n}$ is the variance of the interaction between indicator $x_{i}$ and indicator *n* − 1.The second-order $M_{\mathrm{ij}}$, the third-order $M_{\mathrm{ijk}}$ and the higher-order sensitivity index $M_{\mathrm{ij}\ldots n}$ of the index $x_{i}$ coupled with other indexes can be defined as:(7)$$M_{i}=\frac{V_{i}}{V},M_{ij}=\frac{V_{ij}}{V},M_{ijk}=\frac{V_{ijk}}{V},M_{ij\ldots n}=\frac{V_{ijk\ldots n}}{V}$$The sum of sensitivity for each indicator is:(8)$$M_{mi}=M_{i}+M_{ij}+M_{ijk}+\ldots +M_{ij\ldots n}$$The sensitivities of the coupled indicators were used to calculate weights, and the sensitivity of the first index $M_{\mathrm{mi}}$ normalized weight value $E_{i}$. The weight calculation formula can be expressed as:(9)$$W_{j}^{\prime \prime }{\;=M}_{\mathrm{mi}}/\sum_{i=1}^{n}M_{\mathrm{mi}}$$Fourth item:Index subjective and objective combination weight calculation:(10)$$w_{j}=\frac{W_{j}^{\prime }*W_{j}^{\prime \prime }}{\sum_{j=1}^{n}W_{j}^{\prime }*W_{j}^{\prime \prime }}$$
where $w_{j}$ is the index combination weight value; n is the number of indicators; $W_{j}^{\prime }$ prime is the weight result obtained by three-scale AHP; and $W_{j}^{\prime \prime }$ prime obtains weight results for the EFAST algorithm.

#### 2.4.2. Improved TOPSIS Model

In this paper, the weighted Mahalanobis distance replaced the Euclidean distance of the original TOPSIS model, and the grey correlation analysis was introduced to improve the TOPSIS model. Based on the original TOPSIS model, the improved TOPSIS model comprehensively considered the correlation between various indicators and intuitively showed the nonlinear relationship between sequences. The calculation process is as follows: criterion layer: m and n.


First item:The standardized index matrix $A_{m\times n}$ was established: the matrix $A_{m\times n}$ comprised the standardized index data $x_{\mathrm{ij}}$, where $x_{\mathrm{ij}}$ is the standardized value of the sample j and index i: *m* and *n* are the criterion number of the criterion layer and the index number of the index layer, respectively.Second item:Determine the positive ideal solution $Y_{j}^{+}$ and the negative ideal solution $Y_{j}^{-}$:(11)$$\left\{\begin{array}{c} Y_{j}^{+}{=\max\{Y}_{1j},Y_{2j},\ldots {,Y}_{\mathrm{nj}}\}\\ Y_{j}^{-}{=\min\{Y}_{1j}{,Y}_{2j},\ldots {,Y}_{\mathrm{nj}}\} \end{array}\right.\mathrm{Positive}\mathrm{indexes}$$
(12)$$\left\{\begin{array}{c} Y_{j}^{+}{=\min\{Y}_{1j}{,Y}_{2j},\ldots {,Y}_{\mathrm{nj}}\}\\ Y_{j}^{-}{=\max\{Y}_{1j}{,Y}_{2j},\ldots {,Y}_{\mathrm{nj}}\} \end{array}\right.\mathrm{Negative}\mathrm{indexes}$$Third item:Establishment of Weighted Norm Matrix ${P=(p}_{\mathrm{ij}}{)}_{m\times n}=\left(\begin{array}{ccc} \begin{array}{cc} \begin{array}{c} p_{11}\\ p_{21} \end{array} & \begin{array}{c} p_{12}\\ p_{22} \end{array} \end{array} & \cdots & \begin{array}{c} p_{1n}\\ p_{2n} \end{array}\\ \vdots & \ddots & \vdots \\ \begin{array}{cc} p_{m1} & p_{m2} \end{array} & \cdots & p_{\mathrm{mn}} \end{array}\right)$, where m is the number of evaluation schemes; n is the number of indicators in each evaluation scheme; $p_{\mathrm{ij}}$ is the value multiplied by the corresponding weight value after the standardized transformation.Fourth item:Calculation of weighted Mahalanobis distance. The formula is:(13)$$\left\{\begin{array}{c} d_{i}^{+}=\sqrt{{{(Y}_{\mathrm{ij}}{-Y}^{+})}^{T}w^{T}\sum^{-1}{w(Y}_{\mathrm{ij}}{-Y}^{+})}\\ d_{i}^{-}=\sqrt{{{(Y}_{\mathrm{ij}}{-Y}^{-})}^{T}w^{T}\sum^{-1}{w(Y}_{\mathrm{ij}}{-Y}^{-})} \end{array}\right.$$
where $w=\mathrm{diag}(\sqrt{w_{1}},\sqrt{w_{2}},\ldots ,\sqrt{w_{n}})$; ∑ is the covariance matrix.Fifth item:Calculation of grey correlation degree:(14)$$\left\{\begin{array}{c} r_{ij}^{+}=\frac{\min_{i}\min_{j}\left|v_{j}^{+}-v_{ij}\right|+\rho \max_{i}\max_{j}\left|v_{j}^{+}-v_{ij}\right|}{\left|v_{j}^{+}-v_{ij}\right|+\rho \max_{i}\max_{j}\left|v_{j}^{+}-v_{ij}\right|}\\ r_{ij}^{-}=\frac{\min_{i}\min_{j}\left|v_{j}^{-}-v_{ij}\right|+\rho \max_{i}\max_{j}\left|v_{j}^{-}-v_{ij}\right|}{\left|v_{j}^{-}-v_{ij}\right|+\rho \max_{i}\max_{j}\left|v_{j}^{-}-v_{ij}\right|}\\ r_{i}^{+}=\frac{1}{n}\sum_{j=1}^{n}r_{ij}^{+}\\ r_{i}^{-}=\frac{1}{n}\sum_{j=1}^{n}r_{ij}^{-} \end{array}\right.$$
where ρ = 0.5, $r_{i}^{+}$ is the grey correlation degree between the evaluation scheme and positive ideal solution, $r_{i}^{-}$ is the grey correlation degree between the evaluation scheme and negative ideal solution.Sixth item:Dimensionless processing weighted Mahalanobis distance $d_{i}^{+}$, $d_{i}^{-}$, and grey correlation $r_{i}^{+}$, $r_{i}^{-}$:(15)$$\left\{\begin{array}{c} D_{i}^{+}{=d}_{i}^{+}{/\max}_{i}d_{i}^{+}\\ D_{i}^{-}{=d}_{i}^{-}{/\max}_{i}d_{i}^{-}\\ \begin{array}{c} R_{i}^{+}{=r}_{i}^{+}{/\max}_{i}r_{i}^{+}\\ R_{i}^{-}{=r}_{i}^{-}{/\max}_{i}r_{i}^{-} \end{array} \end{array}\right.$$Seventh item:Calculates the relative paste progress:(16)$$\left\{\begin{array}{c} H_{i}^{+}{=a}_{1}D_{i}^{-}{+\beta }_{1}R_{i}^{+}\\ H_{i}^{-}{=a}_{2}D_{i}^{+}{+\beta }_{2}R_{i}^{-} \end{array}\right.$$
$$C_{i}{\;=H}_{i}^{+}{/(H}_{i}^{+}{+H}_{i}^{-})$$
where $a_{1}{+a}_{2}=1$, $\beta_{1}{+\beta }_{2}=1$; $D_{i}^{-}$, $R_{i}^{+}$, $D_{i}^{+}$, $R_{i}^{-},H_{i}^{+},H_{i}^{-}$ all reflect the distance between each evaluation object and the ideal solution; $C_{i}$ is the relative progress, and the larger the value is, the higher the water cycle health level is, and vice versa.


#### 2.4.3. Obstacle Factor Analysis


First item:Calculation of factor contribution $F_{j}$ for evaluation j:(17)$$F_{j}{\;=w}_{i}^{*}{\times w}_{i}$$
where $w_{i}^{*}$ is the weight value of the criterion layer corresponding to the index.Second item:Calculation of deviation $I_{j}$:(18)$$I_{j}{\;=1-x}_{\mathrm{ij}}$$Third item:Calculation of barriers to evaluation indicators $P_{j}$:(19)$$P_{j}=\frac{F_{j}I_{j}}{\sum_{j=1}^{n}F_{j}I_{j}}$$


The value of $C_{i}$ represents the degree of water cycle health, which is between 0 and 1. A higher $C_{i}$ value indicates a higher water cycle health rating, whereas a lower $C_{i}$ value indicates a lower water cycle health rating. Based on the existing research results and related data, water cycle health was divided into five grades, as shown in Table 3. The flow chart of water cycle health evaluation is shown in Figure 4.

## 3. Results and Discussion

### 3.1. Weight Calculation Results

The subjective weight of the evaluation index was calculated using a three-scale AHP. After calculation,$C_{R}\;=0.05<1$; therefore, the consistency test was passed. The objective weights of evaluation indexes in different years were determined by the EFAST algorithm. The minimum information entropy principle combines the subjective and objective weight values. Finally, the combined weight value of each evaluation index was calculated. The weight values of four criteria layers and 19 indicators of the evaluation system are shown in Figure 5.

### 3.2. Evaluation of Index Layer

The health state for the index layer is shown in Figure 6 according to the threshold of each indicator. According to evaluation results of the index layer, most indicators were developing towards a healthy trend annually.

Among them, the indicators $D_{2},D_{3},B_{4}$ were healthy in the past decade, showing that the sewage treatment rate in Zhengzhou was high, the industry’s water consumption per unit of value (CNY 10,000) was reasonable, and the safety of drinking water sources was high. $A_{1}{,A}_{5}{,B}_{3}$ were opposing indicators and have been in sub-disease and morbid states for a long time in the past decade. The health level of $B_{3}$ was poor, and the health score was only 1.0. Apart from the general health level in 2016, $A_{5}$ was sub-morbid for nine years, and the score was only 2.0. The health level of $A_{1}$ improved in the past three years, but it was still sub-morbid. These three indicators indicated that Zhengzhou had to improve the drainage pipeline density, increase the water consumption of the ecological environment, and draw attention to reservoirs’ annual water storage change. $D_{5}{,D}_{6},A_{4}$ status continued to improve and tended to be healthy. $A_{2}$ experienced three consecutive years of health from 2011 to 2013, but the next seven years were sub-health, showing that the protection of water resources in Zhengzhou City was increased and resulted in improved water ecological functions such as river and lake storage capacity and water function area compliance rates. $C_{3}$ was morbid in 2021 due to the once-in-a-millennium heavy rainfall in Zhengzhou on 20 July 2021, showing that the precipitation in Zhengzhou was unstable. Therefore, the precipitation trend in Zhengzhou should be better monitored. If the precipitation was excessive, there should be a timely warning to avoid the adverse effects of jade. The health status of $C_{4}$ was volatile, showing the dependence of Zhengzhou on groundwater and the sensitivity of indicators.

### 3.3. Factors Analysis of Water Circulation Health Disorders

According to Formulas (17)–(19), the obstacle degrees of 19 evaluation indexes in Zhengzhou City from 2011 to 2021 were calculated. Six indexes with large obstacle degrees were selected for statistical analysis to accurately understand the distribution of obstacle degrees. The calculation results are shown in Table 4. The greatest obstacle factor was the ecological environment’s water consumption ($A_{1}$), and the average obstacle degree was 19.65. The main obstacle factor of the minimum obstacle degree was the green coverage rate in a built-up area ($A_{2}$); the average obstacle degree was 7.3.

Table 4 shows eco-environmental water demand ($A_{1}$), the proportion of river length above class III water quality ($B_{3}$), the industry’s water consumption per unit of value (CNY 10,000) ($D_{2}$), water resources amount ($C_{1}$), effective utilization coefficient of irrigation water ($D_{1}$), and the green coverage rate of the built district ($A_{2}$) when the obstacle degree was high. The six factors with a higher obstacle degree were the main influencing factors. $D_{1}$ and $D_{2}$ belonged to the water utility criterion layer, $C_{1}$ belonged to the water abundance criterion layer, $A_{1}$ and $A_{2}$ belonged to the water ecology criterion layer, and $B_{3}$ belonged to the water quality criterion layer.

In order to visually display the obstacle degree change of the six main obstacle factors, the trend was plotted as a broken line, as shown in Figure 7.

From 2011 to 2021, the eco-environmental water demand ($A_{1}$), the green coverage rate of built districts ($A_{2}$), the water resources amount ($C_{1}$), and the industry’s water consumption per unit of value (CNY 10,000) ($D_{2}$) showed an increasing trend of obstacle degree annually. Therefore, Zhengzhou should increase ecological environment water consumption and improve the green coverage of built-up areas. In residents’ daily life, there should be more water-saving instruments to improve the water resources shortage in Zhengzhou City. In terms of industrial structure, the production mode should be improved to reduce the industry’s water consumption per unit of value (CNY 10,000).

### 3.4. Evaluation of Target Layer

Through the value of each index and the corresponding weight, we established the improved TOPSIS model using Formulas (1)–(16), and the relative progress of Zhengzhou City from 2011 to 2021. The specific results are shown in Table 5.

From 2011 to 2021, the overall health status of the water cycle in Zhengzhou City developed a healthy trend. Due to the low eco-environmental water consumption from 2011 to 2012, the compliance rate of the water function zones was low, and the health level of the water cycle in Zhengzhou City from 2011 to 2012 was grade IV. From 2020 to 2021, the water cycle’s health degree in Zhengzhou City decreased mainly due to the rare heavy rainfall weather on 20 July 2021; the annual average precipitation rose to more than 1000 mm, resulting in severe urban waterlogging, slightly lowering the water cycle’s health degree in 2021 compared with 2020. In recent years, with the implementation of the most stringent water resources management system by the government and the strengthening of natural environmental protection, the water cycle health of Zhengzhou City has developed healthily and steadily.

Based on our analysis of the evaluation results, the countermeasures to improve Zhengzhou City’s water cycle health are as follows:(1)Continue to implement the most stringent water resources management system. In terms of natural environmental protection, we must increase the ecological protection of rivers and lakes and reduce groundwater exploitation.(2)The precipitation in Zhengzhou City is higher in summer; therefore, reservoir water storage and other water storage works should be conducted for subsequent use. If the precipitation is excessive, the reservoir should be discharged to avoid danger.(3)Zhengzhou’s advantages as a transportation hub should be considered, increasing idea exchange, and continuously exploring new ways to develop green ecology.(4)Government departments should improve the temporary emergency response capacity for natural disasters, further improving the flow capacity of urban drainage channels, and accelerating the establishment of safety monitoring facilities for small and medium-sized reservoirs.

### 3.5. Method Validity Test

We selected two conventional assessment methods, the FCE method and the traditional TOPSIS model, to compare our evaluation results to the improved TOPSIS model and verify the effectiveness of the improved TOPSIS model. The results are shown in Table 6.

The evaluation results calculated by the three methods were consistent. There were some differences between the traditional TOPSIS model and the improved TOPSIS model in the evaluation results from 2013 to 2015 and 2021 because, in the improved TOPSIS model, the traditional Euclidean distance was replaced by the weighted Mahalanobis distance and the grey correlation analysis degree. As a result, the correlation degree between some indexes was comprehensively considered, and the position and form indicated the health grades of each year. The FCE method comprehensively considers the subjective opinions of experts and has strong subjectivity [20], whereas the improved TOPSIS method does not need subjective judgment and reflects the relative progress of each evaluation scheme from the original data. The improved TOPSIS model is applicable and scientific in water cycle health evaluation.

## 4. Conclusions

The water cycle plays an important role in promoting the sustainable development of nature and human society. The evaluation and definition of a healthy water cycle relate to the development direction of society, which are of great research significance. In this paper, we defined the concept of a healthy water cycle from several perspectives, and an index system was established based on the natural and social attributes of the water cycle. We analysed and compared the evolution of the water cycle in Zhengzhou City from 2011 to 2021 using the improved TOPSIS model and TOPSIS model and FCE. Our research led to the following conclusions.(1)In recent years, the health degree of the water cycle in Zhengzhou City was between general (level III) and sub-health (level II), and the health score is increasing annually. It reached the sub-health state in 2020, and the relative progress was the highest in the past decade. However, it is still necessary to improve the ecological environment water consumption and increase the proportion of river length above class III water quality to improve the water resources ecology and quality.(2)In the health evaluation system of Zhengzhou City’s water cycle, the obstacles to ecological environment water consumption, built-up area greening coverage rate, total water resources, and the industry’s water consumption per unit of value (CNY 10,000) increased annually. The evaluation results demonstrated that increasing human protection of the ecological environment and improving water-saving awareness can improve the water cycle’s health level.(3)The three methods were highly consistent in Zhengzhou’s water cycle health evaluation results; however, the corresponding health grades were different. The FCE method has strong subjectivity. The Euclidean distance has less consideration for the correlation of some indicators. The improved TOPSIS model uses weighted Mahalanobis distance and grey relational analysis to replace the Euclidean distance, reducing subjective judgment and comprehensively considering the correlation between some indicators. The improved TOPSIS model is more scientific.(4)In this paper, the study area was only Zhengzhou City. In future research, multiple research areas can be selected for comparative study to better judge the applicability of different methods.

## Figures and Tables

**Figure 1 ijerph-19-10552-f001:**
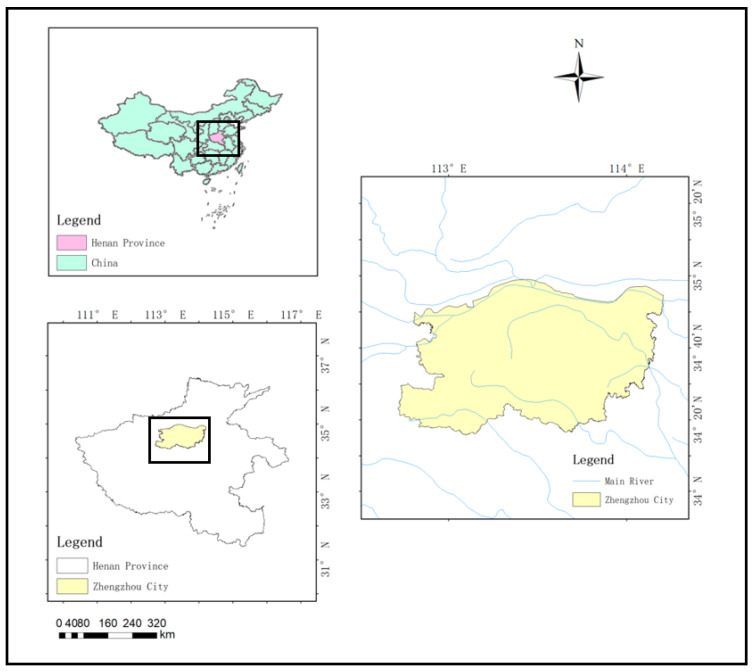
Location of Zhengzhou city.

**Figure 2 ijerph-19-10552-f002:**
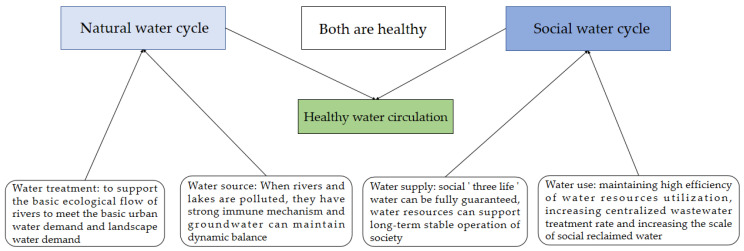
Schematic diagram of urban water cycle health.

**Figure 3 ijerph-19-10552-f003:**
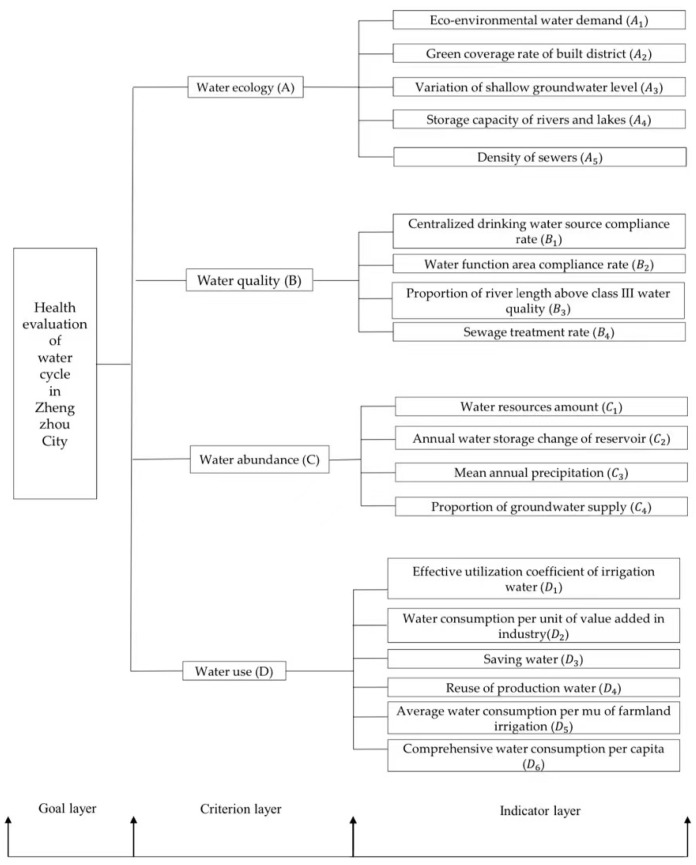
Schematic diagram of the indicator system.

**Figure 4 ijerph-19-10552-f004:**
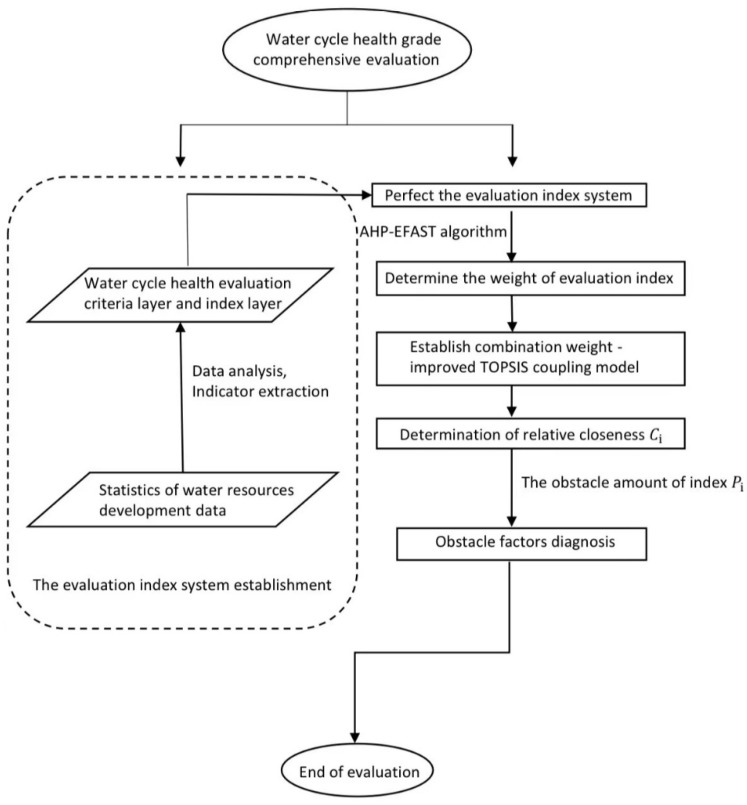
Flow chart of water cycle health assessment.

**Figure 5 ijerph-19-10552-f005:**
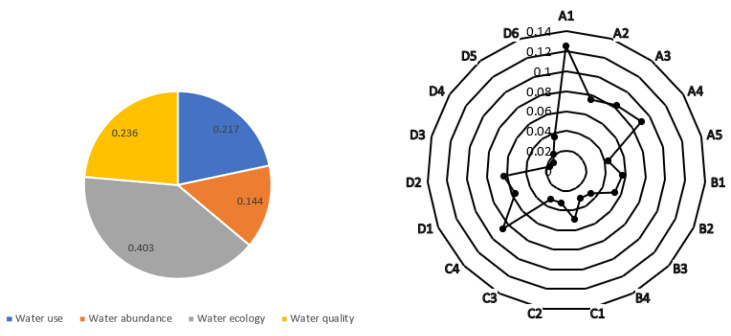
Weight graphs of the criterion and indicator layers.

**Figure 6 ijerph-19-10552-f006:**
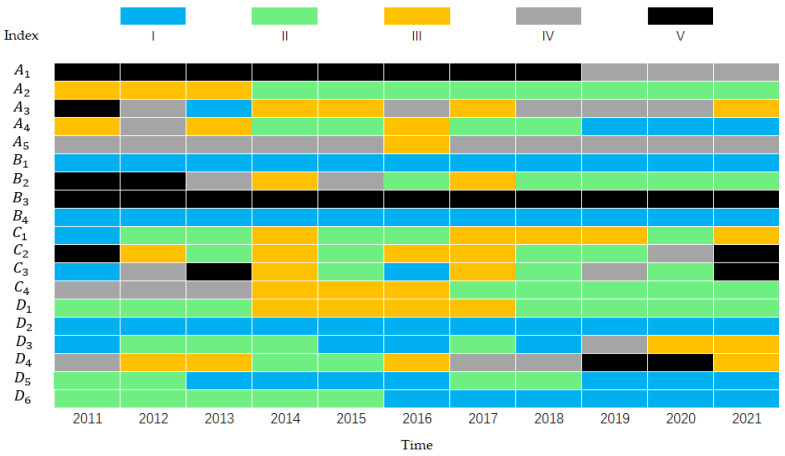
Health states of the evaluation indices in 2011–2021.

**Figure 7 ijerph-19-10552-f007:**
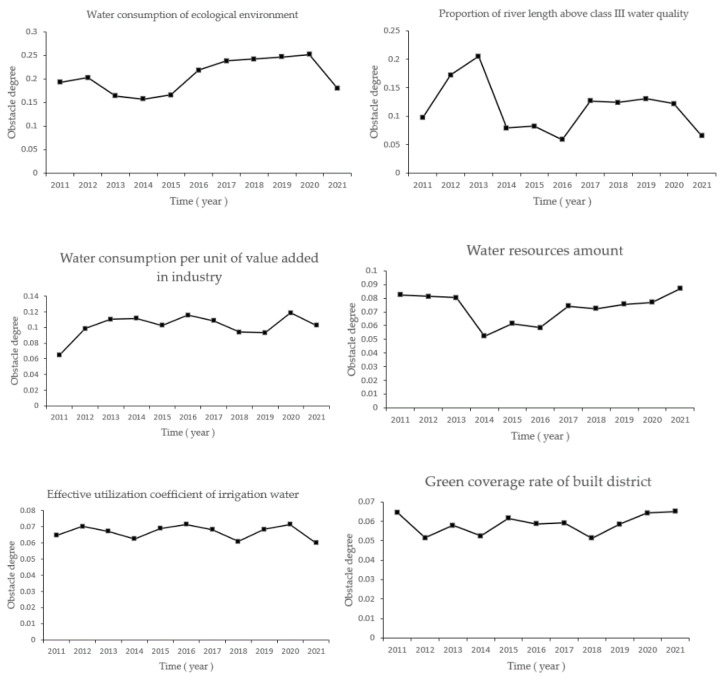
Obstacles of Main Indicators in Zhengzhou City, 2011–2021.

**Table 1 ijerph-19-10552-t001:** Health evaluation of water cycle threshold.

Threshold and Properties of Evaluation Index							
Criterion Layer	Indicator Layer	I	II	III	IV	V	Attribute
		5	(5, 4]	(4, 3]	(3, 2]	(2, 1]	
	A_1_/m^3^ × 10^8^	[25, 20)	[20, 15)	[15, 10)	[10, 5)	[5, 0)	naturality
	A_2_/%	[100, 50)	[50, 40)	[40, 30)	[30, 20)	[20, 10)	nature—sociality
Water ecology	A_3_/m	≤−2	(−2,1]	(1,0]	(0,1]	>1	naturality
	A_4_	strong	relatively strong	general	weaker	feebleness	naturality
	A_5_/km·km^−2^	>15	[15, 12)	[12, 9)	[9, 6)	≤6	Sociality
	B_1_/%	100	(100, 95]	(95, 90]	(90, 80]	<80	nature—sociality
Water quality	B_2_/%	[100, 90)	[90, 60)	[60, 40)	[40, 20)	[20, 0)	nature—sociality
	B_3_/%	100	(100, 95]	(95, 90]	(90, 80]	<80	nature—sociality
	B_4_/%	[100, 95)	[95, 90)	[90, 85)	[85, 80)	≤80	Sociality
	C_1_/m^3^ × 10^8^	>10	[10, 8)	[8, 6)	[6, 4)	≤4	naturality
Water abundance	C_2_/m^3^ × 10^8^	0	(0, 0.05]	(0.05, 0.1]	(0.1, 0.15]	(0.15, 0.2]	sociality
	C_3_/mm	[650, 600)	[600, 550)	[550, 500)	[500, 450)	≤450	naturality
	C_4_/%	[10, 25)	[25, 40)	[40, 55)	[55, 70)	[70, 100)	Naturality
	D_1_	[0.85, 0.75)	[0.75, 0.65)	[0.65, 0.55)	[0.55, 0.45)	[0.45, 0)	sociality
	D_2_/m^3^	[10, 25)	[25, 50)	[50, 100)	[100, 150)	≤150	sociality
Water use	D_3_/m^3^ × 10^8^	>5	[5, 4)	[4, 3)	[3, 2)	≤2	sociality
	D_4_/m^3^ × 10^8^	≥15	[15, 13)	[13, 11)	[11, 9)	≤9	sociality
	D_5_/m^3^	≥150	(150, 100]	(100, 80]	(80, 50]	≤50	nature—sociality
	D_6_/m^3^	>200	[200, 150)	[150, 90)	[90, 50)	≤50	sociality

**Table 2 ijerph-19-10552-t002:** Values of R.I. in judgment matrix.

Order	1 or 2	3	4	5	6	7	8	9
R.I.	0	0.58	0.9	1.12	1.24	1.32	1.41	1.45

**Table 3 ijerph-19-10552-t003:** Classification standard of water cycle health [19].

$C_{i}.$	$\left[0,0.352\right)$	$\left[0.352,0.418\right)$	$\left[0.418,0.525\right)$	$\left[0.525,0.65\right)$	$\left[0.65,1\right)$
Water cycle health grade	V	IV	III	II	I

**Table 4 ijerph-19-10552-t004:** Barrier Factor Analysis for the evaluation index 2011–2021.

Particular Year	1	2	3	4	5	6
2021	$C_{3}$/23.01	$A_{1}$/18.02	$D_{2}$/10.27	$C_{1}$/8.7	$B_{3}$/6.53	$A_{2}$/6.51
2020	$A_{1}$/25.18	$B_{3}$/12.16	$D_{2}$/11.89	$C_{1}$/7.7	$D_{1}$/7.15	$A_{2}$/6.42
2019	$A_{1}$/24.69	$B_{3}$/13.08	$D_{2}$/9.32	$C_{1}$/7.56	$D_{1}$/6.84	$A_{2}$/5.85
2018	$A_{1}$/24.25	$B_{3}$/12.4	$D_{2}$/9.4	$C_{1}$/7.24	$D_{1}$/6.08	$A_{2}$/5.14
2017	$A_{1}$/23.84	$B_{3}$/12.65	$D_{2}$/10.85	$C_{1}$/7.42	$D_{1}$/6.82	$A_{2}$/5.92
2016	$A_{1}$/21.85	$B_{3}$/13.28	$D_{2}$/11.56	$D_{5}$/8.56	$D_{1}$/7.15	$A_{2}$/5.86
2015	$B_{2}$/21.18	$A_{1}$/16.6	$D_{2}$/10.28	$B_{3}$/8.24	$D_{1}$/6.9	$A_{2}$/6.15
2014	$B_{2}$/20.89	$A_{1}$/15.76	$D_{2}$/11.15	$B_{3}$/7.89	$D_{1}$/6.25	$A_{2}$/5.25
2013	$B_{3}$/20.52	$A_{1}$/16.4	$D_{2}$/11.06	$C_{1}$/8.05	$D_{1}$/6.72	$A_{2}$/5.78
2012	$A_{1}$/20.5	$B_{3}$/17.21	$D_{2}$/9.85	$C_{1}$/8.13	$D_{1}$/7.02	$A_{2}$/5.15
2011	$A_{1}$/19.23	$C_{3}$/18.26	$B_{3}$/9.74	$C_{1}$/8.25	$B_{2}$/7.18	$D_{2}$/6.47

**Table 5 ijerph-19-10552-t005:** $C_{i}$ calculation results of Zhengzhou relative patch schedule.

Particular Year	$d_{i}^{+}$	$d_{i}^{-}$	$r_{i}^{+}$	$r_{i}^{-}$	$H_{i}^{+}$	$H_{i}^{-}$	$C_{i}$	Health Level
2011	0.042	0.061	0.715	0.892	0.686	1.052	0.394	IV
2012	0.042	0.062	0.734	0.878	0.684	1.049	0.395	IV
2013	0.038	0.067	0.747	0.874	0.726	1.014	0.417	III
2014	0.035	0.069	0.75	0.862	0.755	0.976	0.436	III
2015	0.032	0.072	0.75	0.851	0.756	0.916	0.452	III
2016	0.031	0.074	0.756	0.834	0.772	0.885	0.466	III
2017	0.029	0.078	0.772	0.816	0.786	0.853	0.48	III
2018	0.025	0.084	0.806	0.797	0.796	0.785	0.503	III
2019	0.023	0.095	0.826	0.786	0.827	0.704	0.54	II
2020	0.019	0.102	0.841	0.765	0.892	0.681	0.567	II
2021	0.018	0.108	0.915	0.762	0.927	0.658	0.523	III

**Table 6 ijerph-19-10552-t006:** Comparison of evaluation results.

Particular Year	FCE	TOPSIS Model	Improved TOPSIS Model
2011	IV	IV	IV
2012	IV	IV	IV
2013	III	IV	III
2014	III	IV	III
2015	III	IV	III
2016	III	III	III
2017	III	III	III
2018	III	III	III
2019	II	II	II
2020	II	II	II
2021	III	II	III

## Data Availability

Not applicable.

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
