# Peer review of "Research on Evaluation Method for Urban Water Circulation Health and Related Applications: A Case Study of Zhengzhou City, Henan Province"

_ijerph, 2022, doi:10.3390/ijerph191710552_

Round 1

Reviewer 1 Report

The poor sentence structure and misuse of words make this manuscript impossible to review. Extensive editing and rewriting are necessary before resubmission.

Author Response

Thank you for your valuable comments on my paper.We have modified the paper according to your opinion.For the language changes, we used the language polish feature of MDPI.Please refer to the attachment for modification details.Thank you again for your valuable comments on our paper.

Reviewer 2 Report

In this paper, the quantitative evaluation method is used to dynamically investigate the water cycle health in Zhengzhou city. The data source of the paper is reliable. In addition, the paper innovatively combines AHP and EFAST methods to make the evaluation results more convincing. Overall, the research is well executed, and the results shed insights for government authorities to enhance related policies. However, I have the following comments for the authors to consider in revising their manuscript:

(1) The time is inconsistent. The year of the data used in the paper should be 2010-2020, not 2011-2021. Authors need to make adjustments to some expressions, including abstracts and full texts.

(2) All variables need to be in italics, M in lines 192 and 198, etc.

(3) It is recommended to use Visio or professional software for pictures 2, 3, and 4 to increase the clarity of the pictures. It is a good starting point to use a diagram to show the information contained in Figure 6, but the perspective effect is not ideal. It is recommended to try different graphs or tables.

(4) It is not easy for each paragraph to be too long. For example, the first paragraph of the introduction is too long. It is recommended to express in good paragraphs so that readers can grasp the main point more quickly.

Author Response

Thank you for your valuable comments on my paper.We have modified the paper according to your opinion.For the language changes, we used the language polish feature of MDPI.Please refer to the attachment for modification details.Thank you again for your valuable comments on our paper.

With best regards,

Sincerely Yours,

Jinhang Li

Round 2

Reviewer 1 Report

As a reviewer, it is difficult to get excited about this manuscript. The intent and significance remain obscure, and the analyses seem completed simply to complete them (relatively little purpose). Small editorial issues:

Line 30: braces is the incorrect word

Lines 69-70: First sentence is unnecessary

Author Response

Respected reviewer :

It is an honour to receive your comments again.Thank you very much for your careful review of our paper.Your valuable opinion is an indispensable part of our team to continue to progress.We have carefully analyzed and revised the paper again according to your opinion.Thank you again for your valuable comments on our paper.Please refer to the attachment for modification details.

Response to Reviewer 1 Comments:

Point 1: Line 30: braces is the incorrect word.

Response 1: Line 30: We replaced braces with supports.

Point 2: Lines 69-70: First sentence is unnecessary.

Response 2: We deleted the first sentence.

With best regards,

Sincerely Yours,

Jinhang Li
